Skeletal ossification of Middle Triassic pachypleurosaur Keichousaurus hui (Reptilia: Sauropterygia) revealed by zinc distribution

Wang Yi-nuo 1
Jiang Da-yong djiang@pku.edu.cn 1
Motani Ryosuke 2
Yao Ming-tao 1
Ji Cheng 3
Sun Zuo-yu 1
Zhou Min 1
1 Department of Geology and Geological Museum, Peking University , Beijing , China
2 Department of Earth and Planetary Sciences, University of California, Davis , CA , United States of America
3 State Key Laboratory of Palaeobiology and Stratigraphy, Nanjing Institute of Geology and Palaeontology, Chinese Academy of Sciences , Nanjing , China
Wu Huiting
Electronic publication date: 2025 Jun 18
Publication date: 2025
Volume: 13
Electronic Location ID: e19475
Received 2025 Feb 21; Accepted 2025 Apr 25
Copyright: ©2025 Wang et al.
Copyright year: 2025
Copyright holder: Wang et al.
License: This is an open access article distributed under the terms of the Creative Commons Attribution License, which permits unrestricted use, distribution, reproduction and adaptation in any medium and for any purpose provided that it is properly attributed. For attribution, the original author(s), title, publication source (PeerJ) and either DOI or URL of the article must be cited.
License URL: https://creativecommons.org/licenses/by/4.0/

Keywords: Triassic, Micro-XRF, Pachypleurosaur, Ossification, Zinc

Funding: National Natural Science Foundation of China 41920104001 Fundamental Research Funds for NIGPAS NGBS202303 Youth Innovation Promotion Association of the CAS 2020313 This research was supported by the National Natural Science Foundation of China (Award Numbers: 41920104001), Fundamental Research Funds for NIGPAS (NGBS202303) and the Youth Innovation Promotion Association of the CAS (to C.J., 2020313). The funders had no role in study design, data collection and analysis, decision to publish, or preparation of the manuscript.

==============================
The elemental composition of seven specimens of Keichousaurus hui, a small pachypleurosaur from the Middle Triassic of China, was analyzed via Micro X-ray fluorescence (Micro-XRF). The results indicate that the distribution of zinc (Zn) is notably correlated with the bones of the skeletons and preferentially enriched in certain areas, unlike other bone-enriched elements such as calcium (Ca), phosphorus (P), and strontium (Sr). Based on the distribution of Zn and its chemical properties, Zn is interpreted as a potential indicator of active ossification in K. hui. A comparative analysis of juvenile and subadult specimens reveals distinct patterns of Zn enrichment, reflecting differential bone development across ontogenetic stages. Notably, the Zn distribution in the subadult specimen of K. hui suggests the ossification of the tarsals gradually progresses from the centre to the periphery of each tarsal bone, which is consistent with typical endochondral ossification observed in extant reptiles. Furthermore, by integrating morphological features with the Zn distribution patterns, we infer that pachyostosis in K. hui develops progressively across all stages of growth and development.

Introduction

Keichousaurus hui is a small pachypleurosaur from the Middle Triassic of the eastern Tethyan region (South China) and serves as the index fossil of the Xingyi Fauna (Young, 1958; Lin, Rieppel & Field Museum of Natural History, 1998; Rieppel, 2000; Jiang et al., 2023). Its ontogenesis, sexual dimorphism, allometric growth, reproductive strategies, and diet preferences have been discussed on the basis of abundant, well preserved, and complete specimens from the Zhuganpo Member of Falang Formation (Ladinian, Middle Triassic) at Xingyi, Guizhou, southwestern China (Cheng, Wu & Ji, 2004; Cheng et al., 2009; Fu et al., 2013; Motani et al., 2015; Xue et al., 2015; Liao et al., 2021; Li et al., 2023). Despite these comprehensive investigations, the ossification of K. hui bones remains poorly understood.

Biologically mineralized hard tissues (e.g., bones, teeth, osteoderms) and delicate soft tissues (e.g., feathers, skins) can preserve biochemical residues that provide vital insights into the physiology of extinct taxa (Erickson et al., 2006; Tumarkin-Deratzian, Vann & Dodson, 2006; Edwards et al., 2011; Vitek et al., 2013; Qvarnstrom, Niedzwiedzki & Zigaite, 2016; Lindgren et al., 2018; Su et al., 2025). Among these, hard tissues (e.g., bones) are more likely to retain endogenous chemical signatures than soft tissues, owing to their higher resistance to degradation and diagenetic alteration during fossilization.

Ossification typically occurs through two distinct processes: direct (intramembranous) and indirect (endochondral) ossification, both of which contribute to the formation of normal bone tissue and are regulated by a complex interplay of physiological factors (González, 2019; De Buffrénil & Houssaye, 2021). The trace element, zinc (Zn), has been proven to have profound impacts on bone physiology in extant species (O’Connor et al., 2020). Zn influences bone ossification either directly, acting on nucleation and mineral growth as a divalent cation, or indirectly, as a cofactor for enzymes such as alkaline phosphatase or other metalloenzymes that are integral to the ossification process (Gomez et al., 1999). For example, alkaline phosphatase—synthesized by osteoblasts—plays a crucial role in cartilage mineralization and bone formation. This enzyme requires divalent cations like Zn2+ as cofactors at the active site to function effectively (Coleman, 1992; Gomez et al., 1999).

The presence of endogenous Zn closely linked to ossification has been well-documented in many fossil bones (Anné et al., 2014; Anné et al., 2016; Anné et al., 2018). Zn exhibits remarkable post-mortem preservation and remains stable throughout the fossilization process, as evidenced by studies on extinct species (Shinomiya et al., 1998; Bergmann et al., 2010; Anné et al., 2014; Anné et al., 2017). Research on various fossil materials, such as extinct cave hyenas, archosaurs and Sirenia, has revealed a strong correlation between Zn and fossilized bones, which provides critical insights into ossification processes, even among heterochronous species deposited in diverse burial environments (Anné et al., 2014; Anné et al., 2016; Anné et al., 2018). Consequently, analyzing the distribution of specific elemental markers, such as Zn, can offer valuable information about the physiological state of ossification in extinct organisms.

The application of Micro X-ray fluorescence (Micro-XRF) on fossil surfaces, with its high spatial resolution, enables the visualization of entire fossils and facilitates large-scale scanning. Recent studies utilizing this non-destructive technique on vertebrate fossils and their surrounding matrices have demonstrated its potential not only to complement skeletal anatomy (Li et al., 2020; Schroder et al., 2022; Chen et al., 2023; Liu et al., 2024; Wang et al., 2024), but also to enhance our understanding of the chemical composition in keratinous structures and soft tissues, such as claw sheaths and feathers in Jianianhualong tengi (Li et al., 2020) and the trunk of Mixosaurus panxianensis (Wang et al., 2024). This study aims to apply this method to the skeletal analysis of K. hui in order to elucidate bone development across various ontogenetic stages, including fetal, juvenile, subadult, and adult. The objectives of this paper are threefold: (1) to reveal the elemental distribution (especially Zn) in skeletons of K. hui; (2) to explore the implications of Zn as a biomarker of ossification in K. hui; (3) to visualize the ossification of certain bones at different ontogenetic stages, and compare with those in extant species.

Materials & Methods

The material for this study includes seven specimens of K. hui from Xingyi City, Guizhou, China. Four specimens are deposited in the Geological Museum of Peking University (GMPKU), and three are in the Xingyi National Geopark Museum (XNGM). During the preparation process, all specimens were handled with great care to preserve the original biological signals within the bones and the surrounding matrix. Efforts were made to minimize the use of chemical solvents to avoid potential damage to these signals. The specimens were measured with a pair of digital calipers with a precision of 0.01 mm using the criteria of Sander (1989). Curved elements, such as the neck and trunk, were measured by aligning a cotton thread along the axis and measuring it with a ruler later. Seven morphological features (Table 1), including humerus length (HL), maximum distal width of humerus (MAXDWH), minimum width of humeral shaft (MINWHS), femur length (FeL), snout-vent length (SVL), standard length (SL) and humeral midshaft circumference (HMC), were measured to distinguish their ontogenetic stages and gender following the previous morphology-based classification scheme (Cheng et al., 2009; Qin, Yu & Luo, 2014; Xue et al., 2015). Specifically, the HL/FeL is used to estimate whether these specimens have reached sexual maturity and then gender identification can be quantified in adult individuals based on MAXDWH/MINWHS, HL/FeL and HL/SL (Lin, Rieppel & Field Museum of Natural History, 1998; Cheng et al., 2009) (Table 1). According to the criteria established by Lin, Rieppel & Field Museum of Natural History (1998), adults are defined as those with a HL/FeL ratio greater than 1, while non-adults are characterized by an HL/FeL ratio less than 1 (Table 1). In adult individuals, genders are assigned following the methodology of Cheng et al. (2009). The diagnostic ratios and their corresponding threshold values are as follows: 1.8 for MAXDWH/MINWHS, 1.1 for HL/FeL, and 1.3 for HL/SL (Cheng et al., 2009). Specimens exhibiting at least one ratio exceeding its respective threshold were classified as male (Table 1). Non-adult ontogenetic stages were further subdivided into juvenile (within one year after birth) and subadult (more than one year, but younger than two years) by humeral midshaft circumference (Table 1; Fig. 1). Based on the data of humeral midshaft circumference and age estimated by bone histology from Li et al. (2023), we fitted these data and assigned our specimens to facilitate age division (Fig. 1). However, the humerus of XNGM WS-31-R22 was broken, so some related data are unavailable. GMPKU-P-4317 is a gravid specimen in ventral view whose fetuses cover the four most posterior trunk vertebrae, so the standard length of GMPKU-P-4317 is also unavailable.

Table 1 Measurements of the skeleton in K. hui (mm).

SN	HL	MAXDWH	MINWHS	HMC	FeL	SVL	SL	MAXDWH/ MINWHS	HL/FL	HL/SL	OS	G	
XNGM WS-31-R22	4.45	–	–	–	4.78	45.72	4.37	–	0.93	1.02	–	–	
XNGM WS-30-R43	5.85	1.76	1.09	3.42	6.82	59.54	5.28	1.62	0.86	1.11	juvenile	–	
GMPKU-P-1154(2)	7.40	2.17	1.26	3.96	8.40	88.43	7.43	1.72	0.88	1.00	juvenile	–	
GMPKU-P-4316	11.62	2.93	2.26	7.10	11.94	117.61	10.57	1.30	0.97	1.10	subadult	–	
GMPKU-P-4317	17.52	4.21	3.24	10.17	16.75	167.97	–	1.30	1.05	–	adult	female	
GMPKU-P-4318	22.26	5.27	3.67	11.52	20.67	166.28	17.38	1.44	1.08	1.28	adult	female	
XNGM WS-32-R18	26.44	6.57	3.48	10.92	21.20	178.74	15.82	1.89	1.25	1.67	adult	male	
Notes.

SN specimen number

HL humerus length

MAXDWH maximum distal width of humerus

MINWHS minimum width of humeral shaft

HMC humeral midshaft circumference

FeL femur length

SVL snout-vent length

SL standard length

OS ontogenetic stage

G gender

Figure 1 Regression of the body size (proxy: humeral midshaft circumference) on the age of K. hui (R2:0.86124) (modified based on data from Li et al., 2023), with three specimens superimposed.

The red shaded area represents the 95% confidence interval. The plots of our examined specimens show that XNGM WS-30-R43, GMPKU-P-4316, and XNGM WS-32-R18 are juvenile, subadult and adult, respectively.

Elemental maps of the fossil specimens were obtained using the Bruker M6 Jetstream mobile X-ray fluorescence instrument at the Micro-XRF imaging Lab, China University of Geosciences Beijing. The seven skeletons are dorsoventrally preserved and the instrument was calibrated to mitigate height disparities, ensuring that the variance in height at the four corners did not exceed 2 mm, thereby meeting the requirements for qualitative imaging. Specific mapping parameters, including acquisition conditions, acquisition time per pixel and total measurement time, are in Table S1. Areas of interest, such as the left and right hindlimbs of GMPKU-P-4316, were further investigated by Micro-XRF using higher spatial resolution. The Golden Software Surfer 13 (Golden Software, LLC., Golden, Colorado, USA) was used to analyze the samples’ elemental content data and generate element distribution maps. The XRF mapping is a false-colour image, which only shows the distribution of elements and not their actual concentration.

Results

Results are reported here mainly based on three of the specimens that illustrate ontogenetic differences best, namely XNGM WS-30-R43 (juvenile), GMPKU-P-4316 (subadult), and XNGM WS-32-R18 (adult) (Fig. 1).

Tendency of overall elemental distribution

The elemental distribution in K. hui demonstrates a distinct enrichment of elements typically associated with bone, primarily calcium (Ca), phosphorus (P), and strontium (Sr) (Figs. 2–4, Figs. S1–S4). Conversely, manganese (Mn), sulfur (S), titanium (Ti), and potassium (K) exhibit higher concentration in the surrounding matrix and are relatively depleted in the skeletal structures (Fig. S5). Notably, Zn shows a strong correlation with skeletal elements in some specimens (Figs. 2E, 3E, 4E).

Figure 2 Elemental distribution map and drawing of XNGM WS-30-R43.

(A) Photo of XNGM WS-30-R43. (B–E) Micro-XRF maps (false-color images) of Ca (B), P (C), Sr (D), Zn (E). (F) Drawing of XNGM WS-30-R43 and distribution diagram of Zn. The scale bars are 20 mm. Abbreviations: ti, tibia; fi, fibula; isc, ischium; fe, femur; pub, pubis; ra, radius; ul, ulna; h, humerus; cor, coracoid; sca, scapula; cl, clavicle.

Figure 3 Elemental distribution map and drawing of GMPKU-P-4316.

(A) Photo of GMPKU-P-4316. (B–E) Micro-XRF maps (false-color images) of Ca (B), P (C), Sr (D), Zn (E). (F) Drawing of GMPKU-P-4316 and distribution diagram of Zn. The scale bars are 3 cm. Abbreviations: cl, clavicle; sca, scapula; cor, coracoid; h, humerus; ra, radius; im, intermedium; ul, ulna; ulr, ulnare; ast, astragalus; ti, tibia; pub, pubis; fi, fibula; fe, femur; cal, calcaneum; il, ilium; isc: ischium.

Figure 4 Elemental distribution map and drawing of XNGM WS-32-R18.

(A) Photo of XNGM WS-32-R18. (B–E) Micro-XRF maps (false-color images) of Ca (B), P (C), Sr (D), Zn (E). (F) Drawing of XNGM WS-32-R18 and distribution diagram of Zn. The scale bars are 4 cm. Abbreviations: sca, scapula; h, humerus; ul, ulna; ulr, ulnare; ra, radius; im, intermedium; dc4, fourth distal carpal; pub, pubis; fe, femur; isc, ischium; fi, fibula; ti, tibia; ast, astragalus; cal, calcaneum; dt4, fourth distal tarsal; il, ilium.

The skeletal structures exhibit significantly higher Ca content compared to the surrounding matrix, mineralogically consistent with apatite. However, the contrast in Ca-maps is not sufficiently pronounced to clearly delineate the skeletal outlines (e.g., Figs. 2B, 3B, 4B). P is primarily distributed in skeletons, thereby revealing the skeletal structures (e.g., Figs. 2C, 3C, 4C). Notably, all seven K. hui specimens consistently exhibit distinctly elevated Sr levels throughout the entire skeleton, providing exceptional anatomical clarity (e.g., Figs. 2D, 3D, 4D). The Sr-maps demonstrate a higher contrast compared to the Ca-maps and P-maps, suggesting their superior potential for distinguishing bones from the matrix.

The distribution of Zn in K. hui varies significantly among the seven specimens. In three specimens—XNGM WS-30-R43 (Fig. 2E), GMPKU-P-4316 (Fig. 3E), and XNGM WS-32-R18 (Fig. 4E)—Zn shows marked correlation with skeletal structures, but with distinct distribution patterns in each specimen. In contrast, the remaining four specimens show no elevated Zn levels in the skeletal regions (Figs. S1E, S2E, S3E, S4E). Given this variability, the Zn distribution in the three specimens displaying skeletal enrichment is described in greater detail below.

Elemental distribution in XNGM WS-30-R43, GMPKU-P-4316 and XNGM WS-32-R18

Elemental distribution in XNGM WS-30-R43

XNGM WS-30-R43 is a well-preserved juvenile individual in ventral view (Fig. 2). Zn is primarily concentrated in the skull and along the axial skeleton, particularly in the cervical vertebrae and caudal vertebrae (Figs. 2E, 2F). Notably, Zn enrichment is also observed in the proximal ends and shafts of the dorsal ribs, as well as both the proximal and distal ends of the long bones in the limbs (Fig. 2F).

Elemental distribution in GMPKU-P-4316

GMPKU-P-4316 is a subadult individual exposed in ventral view (Fig. 3). The Zn-map shows that the distribution of Zn is closely related to the skeletal structure, with higher Zn content observed in specific bones such as the parietal, carpals, tarsals, proximal ends and shafts of the dorsal and caudal ribs. Notably, the proximal ends and shafts of dorsal and caudal ribs show distinct Zn enrichment compared to the rest of the ribs, forming a bilaterally symmetrical pattern along the dorsal and caudal vertebrae (Figs. 3E, 3F). Zn distribution around the tarsals is particularly noteworthy, showing a nearly circular pattern consistent with the bone shape (Figs. 5F, 5K, 6J, 6K). According to the complete outline of the tarsals in the light photo and Sr-map (since Sr can clearly delineate bone morphology, we use Sr-maps here to reveal the structure), high Ca and P concentrations are concentrated primarily in the centers of the astragalus and calcaneum (Figs. 5C, 5D, 5H, 5I, 6C, 6D, 6G, 6H). In contrast, Zn distribution is roughly complementary to Ca and P, extending slightly beyond the skeletal outline of the tarsals (Figs. 5L, 5M, 6L, 6M). The Zn-map further reveals the possible morphology of the fourth distal tarsal (dt4). In the left hindlimb, the dt4 is visible in the light photo, Sr-map, and Zn-map, appearing as a subround area between the astragalus and calcaneum (Figs. 5B, 5E, 5F), while being undetectable in the Ca-map or P-map (Figs. 5C, 5D). Notably, this subround Zn-enriched area is larger than those observed in the Sr-map and light photo (Figs. 5B, 5E, 5F). In the right hindlimb, the dt4 is only visible in the Zn-map (Fig. 6J, 6K) and not in other maps (Figs. 6C, 6D, 6E). Furthermore, the Zn content in the metatarsal close to the tarsal gradually diminishes from the proximal towards the distal end in both hindlimbs (Figs. 5F, 5K, 6J, 6K).

Figure 5 Elemental distribution map and drawing of GMPKU-P-4316’s left hindlimb.

(A) The position of left hindlimb. The scale bar is 3 cm. (B) Photo of GMPKU-P-4316’s left hindlimb. The scale bar is 3 mm. (C–F) Micro-XRF maps (false-color images) of Ca (C), P (D), Sr (E), Zn (F). The scale bars are 3 mm. (G) Drawing of GMPKU-P-4316’s left hindlimb. (H–K) The drawing of different elemental distribution, Ca (H), P (I), Sr (J), Zn (K). (L) The distribution of Zn and Ca in GMPKU-P-4316’s tarsal. The scale bar is 1 mm. (M) The distribution of Zn and P in GMPKU-P-4316’s tarsal. The scale bar is 1 mm. (N) The distribution of Zn in GMPKU-P-4316’s astragalus, calcaneum, and the dt4. The scale bar is 0.5 mm. Abbreviations: dt4, fourth distal tarsal; ast, astragalus; cal, calcaneum; ti, tibia; fi, fibula; fe, femur.

Figure 6 Elemental distribution map and drawing of GMPKU-P-4316 right hindlimb.

(A) The position of the right hindlimb. The scale bar is 3 cm. (B) Photo of GMPKU-P-4316’s right hindlimb. The scale bar is 5 mm. (C–E, J) Micro-XRF maps (false-colour images) of Ca (C), P (D), Sr (E), Zn (J). The scale bars are 5 mm. (F) Drawing of GMPKU-P-4316’s right hindlimb. (G–I, K) The drawing of different elemental distribution, Ca (G), P (H), Sr (I), Zn (K). (L) the distribution of Zn and Ca in GMPKU-P-4316’s tarsal. The scale bar is 2 mm. (M) The distribution of Zn and P in GMPKU-P-4316’s tarsal. The scale bar is 2 mm. (N) The distribution of Zn in GMPKU-P-4316’s astragalus, calcaneum, and dt4. The scale bar is 1 mm. Abbreviations: fe, femur; fi, fibula; cal, calcaneum; ti, tibia; ast, astragalus; dt4, fourth distal tarsal.

Figure 7 The ossification pattern in K. hui.

(A) The different bone development patterns in juvenile and subadult individuals. (B) The ossification pattern of tarsus in K. hui.

Elemental distribution of XNGM WS-32-R18

XNGM WS-32-R18 is an adult male in dorsal view, missing part of the tail (Fig. 4). Zn also appears to be enriched in some bones. Interestingly, Zn accumulates at the articular surfaces where each centrum connects with the previous one (Fig. 4E). The proximal regions of the dorsal and caudal ribs exhibit pronounced Zn enrichment relative to the remaining rib segments. And the neural arches of both dorsal and caudal vertebrae also exhibit elevated Zn concentrations. Similarly, the proximal and distal epiphyses of the limb long bones demonstrate significant Zn accumulation. The two dark parts in the trunk region covered by the dorsal ribs and the gastral ribs also show the high level of Zn (Fig. 4E).

Discussion

Preservation mechanisms of Ca, P, Sr, Zn within fossil bones

Changes in the elemental composition of fossil bones can occur during two distinct periods: first, during the organism’s lifetime, and second, during the fossilization process (Piga et al., 2011). Throughout fossilization, the preservation mechanisms of various elements differ due to a complex interplay of chemical, physical, and environmental factors (Parker & Toots, 1970; Pyzalla et al., 2008; Piga et al., 2011). Calcium hydroxyapatite (Ca10(PO4)6(OH)2), the primary inorganic content of bones, has the flexibility of the lattice for replacements at every site within the mineral structure. This flexibility enables the replacement of Ca2+, PO43−, and OH− ions with other ions, facilitating both the uptake and loss of elements within the bones (Miyaji, Kono & Suyama, 2005; Trueman et al., 2008).

Despite some inevitable loss during fossilization, Ca and P are highly tenacious and stable, remaining the predominant constituents in fossil bones (Pan & Fleet, 2002). For instance, in the bones of the Middle Triassic marine reptile Mixosaurus panxianensis, Ca accounts for 68.86 wt% to 71.11 wt%, and P ranges from 6.67 wt % to 8.05 wt % (Wang et al., 2024). A similar pattern is ubiquitous across diverse fossil taxa (Li et al., 2020; Liu et al., 2024), and this trend is also observed in seven specimens of K. hui analyzed in this study (e.g., Figs. 2B–2C, 3B–3C, 4B–4C). The enrichment of Sr in fossils occurs both during the organism’s lifetime and through the fossilization process. Sr concentrations are higher in seawater than that in freshwater, resulting in marine organisms typically exhibiting higher Sr content that is strongly correlated with the Sr levels in their environment (Rosenthal, Eves & Cochran, 1970). Sr incorporates into the bones during fossilization through substitution processes, leading to further enrichment, reflected by the elevated Sr content observed in the seven K.hui specimens (e.g.,Figs. 2D, 3D, 4D) (Gueriau, Jauvion & Mocuta, 2018).

However, the preservation of Zn in fossil bones presents significant challenges, as some degree of Zn loss is inevitable during the fossilization and preparation processes. Firstly, in living organisms, Zn is present in much lower concentrations and is unevenly distributed within bones compared to the abundant endogenous Ca and P (Honda et al., 1984; Goodwin et al., 2007; Wang et al., 2024). Secondly, diagenetic processes, particularly demineralization, are likely responsible for postmortem Zn loss (Dean et al., 2023). Micro-XRF scans of Mixosaurus panxianensis have revealed a pattern of Zn diffusion, with Zn concentrations gradually decreasing from the trunk region toward the surrounding matrix (Wang et al., 2024). Finally, the fossil preparation process can also alter the original distribution of Zn, despite efforts to minimize such impacts. Physical friction during matrix removal can damage the original Zn signal, and the use of solvents such as water or acetone during preparation can lead to Zn loss. Notably, Zn2+ exhibits high solubility in water, increasing the likelihood of its removal during preparation (Mou, 1999). As a result, among the seven K. hui specimens analyzed, only three retained a relatively intact original Zn signal (Figs. 2E; 3E; 4E). The remaining specimens failed to preserve this signal due to factors related to fossilization or preparation.

Origin of Zn in Keichousaurus hui

The robust correlation between Zn intensity and bone morphology in specimens XNGM WS-30-R43, GMPKU-P-4316, and XNGM WS-32-R18 suggests that the observed Zn enrichment in these fossilized bones is endogenous (Figs. 2E, 3E, 4E). This interpretation is supported by several lines of evidence. Firstly, unlike other bone-related elements such as Ca, P, and Sr, which are uniformly distributed throughout the skeletal structures, Zn shows selective accumulation in specific skeletal regions. For example, Zn is concentrated in areas like the periphery of tarsal bones in K. hui (Figs. 5 and 6). Zn exhibits an uneven distribution within the bones of extant and extinct species, predominantly localizing in regions between mineralized and unmineralized zones of osteons, as well as at the ossification front where active calcification occurs (Bradley, Moger & Winlove, 2007; Anné et al., 2014). For instance, synchrotron microfocus X-ray analysis of the forelimbs of one-day-old mice supports this pattern, demonstrating Zn enrichment across all skeletal elements, including both cartilage and bone (Fig. 8D). Peak concentrations are observed at the developing edges of bones, a distribution pattern strikingly similar to that observed in K. hui (Fig. 8D) (Anné et al., 2017). Secondly, Zn content in the surrounding pelitic limestone is markedly lower than that detected in the bones of the three specimens, effectively ruling out exogenous Zn. And there is no evidence to suggest that Zn could have been transferred by aqueous fluids in this system. Mineralogical experiments indicate that substituting exogenous Zn for Ca in hydroxyapatite is relatively challenging (Miyaji, Kono & Suyama, 2005). This contrasts with the physiological incorporation of Zn into bone, suggesting that Zn from groundwater is unlikely to enter bones through substitution processes. Finally, the three Zn-enriched specimens can be compared with the Mixosaurus panxianensis specimen that shows evident Zn loss (Wang et al., 2024), revealing different Zn distribution characteristics: K. hui displays discrete Zn zoning with sharp compositional boundaries (e.g., Fig. 4E), whereas M. panxianensis exhibits diffuse boundaries (Wang et al., 2024). This sharp contrast indicates that the absence of Zn in other skeleton regions is not likely to be attributed to post-depositional loss processes, given that the element loss tends to lead to the diffuse distribution as shown in M. panxianensis.

Figure 8 Comparison of limb ossification between Keichousaurus hui, extant reptile and mammal.

(A, E) The distribution of Zn in GMPKU-P-4316’s carpal and tarsal, respectively. (B, F) The cartilage and bone in the carpal and tarsal of Cyrtodactylus pubisulcus, respectively (modified from Rieppel, 1992). (C, G) The cartilage and bone in the carpal and tarsal of mouse (modified from Patton & Kaufman, 1995). (D) The distribution of Zn in mouse (modified from Anné et al., 2017).

Hence, the Zn enrichment is more likely to be endogenic, i.e., the Zn was accumulated when these K. hui were still alive. The distribution of Zn reflects the original organism’s chemical remains rather than taphonomic artifacts added during fossilization, despite the preservation mechanism remains unclear.

The different ossification patterns in juvenile and subadult individuals

The distribution patterns of Zn, especially the enrichments in the XNGM WS-30-R43 and GMPKU-P-4316 (Figs. 2–3), demonstrate active ossification in K. hui. XNGM WS-30-R43, a juvenile specimen, exhibits higher Zn levels predominantly in the axial skeleton compared to the limbs, suggesting that the axial bones develop more rapidly or earlier than the limb bones at this ontogenetic stage (Fig. 2E). GMPKU-P-4316, a subadult individual, displays high Zn levels in scattered areas such as the carpals, tarsus, parietal, proximal ends of dorsal ribs, and caudal ribs (Fig. 3), suggesting active ossification in these regions during the subadult stage. Particularly, Zn enrichment in the parietal bone indicates that this region was actively ossifying, with the parietal opening gradually closing (Figs. 3E, 3F). This pattern aligns with observations in certain extant reptiles, where closure of the parietal fontanelle is markedly delayed (Maisano, 2001). Similarly, the high Zn content in the ends and shafts of dorsal ribs implies continued ossification in these regions (Figs. 3E, 3F). These findings align with previous descriptions of subadult K. hui by Lin, Rieppel & Field Museum of Natural History (1998) and Qin, Yu & Luo (2014), which highlight rapid ossification of sexually dimorphic limb elements, expansion of dorsal rib ends, and closure of the parietal opening. The observed Zn distribution in GMPKU-P-4316 corroborates these developmental patterns.

In subadult specimens, Zn enrichment is notably absent in the neck and certain cranial regions, with a sharp demarcation line separating these areas from others, suggesting an abrupt rather than gradual transition (Fig. 3E). Microscopic observations revealed that the specimen had been somewhat excessively prepared, potentially resulting in the loss of original Zn signals. Alternatively, Zn may not have been preserved in these regions during fossilization, as discussed earlier, due to the challenges associated with its preservation. Consequently, it cannot be definitively concluded that the neck of subadult individuals lacked Zn enrichment. It is plausible that the neck, like the tail, underwent active ossification, consistent with previous allometric growth analyses (Xue et al., 2015). Unfortunately, this detail is not reflected in the Zn distribution maps, despite meticulous fossil preparation efforts.

The Zn-maps of XNGM WS-30-R43 and GMPKU-P-4316 reveal distinct ontogenetic patterns of ossification, indicating rapid development in both juvenile and subadult individuals (Fig. 7A). In the juvenile period, ossification mainly occurred in the axial bones and the limbs, while the actively ossifying bones in subadult individual are in the limbs (Figs. 2E, 2F, 3E, 3F).

Ossification patterns of tarsus in Keichousaurus hui and comparison with extant species

The ossification pattern of the tarsus in K. hui, as indicated by the distribution of Zn, demonstrates directionality from the centre to the periphery (Fig. 7B). This pattern aligns closely with the highly conserved ossification sequence observed in extant reptiles, which exemplifies typical endochondral ossification (Rieppel, 1991; Maisano, 2001). In this pattern, the ossification center of the tarsal bones initially appears as relatively small regions within larger cartilage primordia and subsequently expands radially from the centre to the periphery of each tarsal element (Hemo, Gigi & Wientroub, 2019; Maisano, 2000; Maisano, 2002). In Cyrtodactylus pubisulcus (Gekkonidae), the ossification centers similarly originate within the cartilage primordia and undergo gradual expansion over time, and the tarsal ossification sequence begins with the astragalus, followed by the calcaneum, and then the dt4. In terms of size, the tarsal elements follow the order astragalus >calcaneum >dt4 (Rieppel, 1992) (Fig. 8F).

Similarly, in early postnatal laboratory mice, ossification centre is also in the cartilage primordia and gradually expanded (Patton & Kaufman, 1995) (Fig. 8G). Notably, in the astragalus and calcaneum of GMPKU-P-4316, Zn exhibits higher concentrations in the peripheries compared to the centres (Figs. 5–6), suggesting that the peripheries were actively undergoing ossification while the centres may have already completed ossification. Additionally, the concentrations of Ca and P in the centres of the astragalus and calcaneum exceed those in the peripheries, complementing the distribution pattern of Zn (Figs. 5L, 5M, 6L, 6M). This further supports the inference that the degree of calcification is more advanced in the centers than in the peripheries. A partial extension of Zn is observed beyond the supporting bony elements, distributed along the sub-round bone margin in hindlimb (Fig. 5N, 6N), which may represent biological residues from the bone development process. Furthermore, the right hindlimb of GMPKU-P-4316 displays a subcircular area in the Zn-map (Fig. 6J) between the astragalus and calcaneum, which is absent in the light photo, Ca-map, P-map, and Sr-map (Figs. 6B–6E). This subcircular region in the Zn map may correspond to the biological remains of the dt4, which appears not to have ossified yet. By comparing the elemental distributions between the centers and peripheries, we can gain deeper insights into the ossification process of the tarsus in K. hui. This analysis supports the conclusion that the ossification pattern of the tarsus in K. hui follows a gradual progression from the center to the periphery (Fig. 7B).

According to the element maps, the astragalus exhibits a larger relative area enriched in Ca and P compared to the calcaneum, indicating that the extent of ossification is more advanced in the astragalus than in the calcaneum (Figs. 5L, 5M, 5N, 6L, 6M, 6N). Morphological observations further reveal that the irregular shape of the calcaneum suggests incomplete ossification. Therefore, the sequence of tarsal ossification in K. hui can be inferred as astragalus, calcaneum, and dt4, which aligns with the ossification pattern observed in certain members of Gekkonidae (Rieppel, 1992).

The ossification of ribs in Keichousaurus hui

Pachyostosis, a non-pathological bone hypertrophy, is observed in a wide array of taxa adapted to marine environments, irrespective of their phylogenetic relationships (Houssaye, 2009). The increase in skeletal mass plays the functional role of ballast for buoyancy control and hydrostatic regulation of body trim (Ricqlès & de Buffrénil, 2001; Houssaye, 2009). The Middle Triassic pachypleurosaurs, which inhabited shallow epicontinental marine habitats and intraplatform basins, display pachyostosis in most elements of the presacral axial skeleton (Sues & Carroll, 1985; Rieppel & Kebang, 1995; O’Keefe, Rieppel & Sander, 1999). In K. hui, pachyostosis is particularly evident in the dorsal region, where the ribs and vertebrae are thickened (Young, 1958).

In the juvenile individual (XNGM WS-30-R43), the shafts of dorsal ribs exhibit elevated Zn content (Figs. 2E, 2F), suggesting active ossification in these regions. Similarly, the subadult individual (GMPKU-P-4294) shows Zn enrichment in the shafts of dorsal ribs, though the areas with high Zn content are less extensive compared to the juvenile (Figs. 3E, 3F). Notably, pachyostosis is not apparent in either the juvenile or subadult specimens (Figs. 2A–2D; 3A–3D). The adult individual exhibits significant Zn enrichment in the rib shafts, suggesting that ossification is still ongoing, while pachyostosis is clearly evident (Fig. 4E).

Previous morphological studies have indicated that pachyostosis is absent in K. hui embryos (Young, 1958). Our observations of juvenile and subadult individuals also show no clear signs of pachyostosis, but the pachyostosis is apparent in adult individual. In addition, the distribution of Zn highlights active ossification in the shafts of the dorsal ribs in juvenile, subadult and adult individuals. Integrating these morphological features with Zn distribution patterns, we infer that pachyostosis in K. hui develops progressively from embryonic stages to adulthood.

The advantage of Zn revealing the ossification information in fossils

Zn, preserved in situ over deep time, enables the identification of active ossification sites related to ontogenetic stages, particularly in non-adult extinct organisms. Moreover, the macroscopic enrichment of Zn can elucidate the ossification pattern of different ontogenetic individuals, providing more information about growth pattern.

Comparative anatomy and bone histology techniques have been used to reconstruct the ossification patterns in fossil marine reptiles (Klein & Sander, 2008; Scheyer, Klein & Sander, 2010). However, comparative anatomy as direct investigation is usually hampered by preservative conditions and the number of specimens (Horner & Weishampel, 1996). Paleohistology which can provide an independent assessment of the age or ontogenetic stage of extinct species based on developmental records in skeletal tissues, is a destructive method and unsuitable for rare fossil specimens (Klein & Sander, 2008). Hence, XRF mapping of Zn presents a novel, non-destructive method for addressing ossification, especially in precious specimens, offering additional insights into ossification patterns in fossils.

Conclusion

This study utilized Micro-XRF scanning to analyze seven Keichousaurus hui specimens at different ontogenetic stages. In addition to the conventional elements, the presence of Zn was identified, indicating active ossification in K. hui. The enrichment of Zn indicates ossification patterns at different ontogenetic stages. Moreover, the ossification process of the tarsal, gradual ossification from the centre to the periphery like extant reptiles, is also revealed by high Zn concentration. In addition, through the integration of morphological characteristics with the distribution of Zn, we deduce that pachyostosis in K. hui develops progressively throughout the organism’s developmental stages. The distribution of Zn revealed by Micro-XRF offers an innovative technique for investigating ossification, especially advantageous for analyzing delicate or irreplaceable specimens, while enhancing our comprehension of ossification processes in fossils.

Supplemental Information

Supplemental Information 1 Elemental distribution map and drawing of XNGM WS-31-R22

(A) XNGM WS-31-R22. (B)-(E) Micro-XRF maps (false-color images) of Ca(B), P(C), Sr(D), Zn(E).

Supplemental Information 2 Elemental distribution map and drawing of GMPKU-P-1154(2)

(A) Photo of GMPKU-P-1154(2). (B)-(E) Micro-XRF maps (false-color images) of Ca(B), P(C), Sr(D), Zn(E).

Supplemental Information 3 Elemental distribution map and drawing of GMPKU-P-4317

(A) Photo of GMPKU-P-4317. (B)-(E) Micro-XRF maps (false-color images) of Ca(B), P(C), Sr(D), Zn(E).

Supplemental Information 4 Elemental distribution map and drawing of GMPKU-P-4318

(A) Photo of GMPKU-P-4318. (B)–(E) Micro-XRF maps (false-color images) of Ca(B), P(C), Sr(D), Zn(E).

Supplemental Information 5 Exogenous elemental distribution map of GMPKU-P-4316

(A)–(D) Micro-XRF maps (false-color images) of Mn(A), S(B), Ti(C), K(D)

Supplemental Information 6 The experiment parameter of Micro-XRF in Keichousaurus hui

We express our sincere gratitude to Tian-fen Hu for specimen preparation (GMPKU series) and Yun-zhong Wang for specimen preparation (XNGM series). Special thanks are extended to Jie Yang and Yan Zhang (China University of Geosciences Beijing) for technical assistance with Micro-XRF scanning. We acknowledge Cindy X. Su (Peking University) for linguistic refinement of the manuscript, and are particularly indebted to Ya-lei Yin, Jun Chai, Jia-chun Li, and Shu-lun Gu (Peking University) for their scholarly insights during conceptual discussions.

Additional Information and Declarations

Competing Interests

Author Contributions

Data Availability

The authors declare there are no competing interests.

Yi-nuo Wang conceived and designed the experiments, performed the experiments, analyzed the data, prepared figures and/or tables, and approved the final draft.

Da-yong Jiang conceived and designed the experiments, prepared figures and/or tables, authored or reviewed drafts of the article, and approved the final draft.

Ryosuke Motani conceived and designed the experiments, prepared figures and/or tables, authored or reviewed drafts of the article, and approved the final draft.

Ming-tao Yao performed the experiments, analyzed the data, authored or reviewed drafts of the article, and approved the final draft.

Cheng Ji conceived and designed the experiments, authored or reviewed drafts of the article, and approved the final draft.

Zuo-yu Sun conceived and designed the experiments, authored or reviewed drafts of the article, and approved the final draft.

Min Zhou analyzed the data, authored or reviewed drafts of the article, and approved the final draft.

The following information was supplied regarding data availability:

The raw data are available in the Supplementary File.

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
