# Peer review of "Skeletal ossification of Middle Triassic pachypleurosaur Keichousaurus hui (Reptilia: Sauropterygia) revealed by zinc distribution"

_PeerJ, doi:10.7717/peerj.19475_

## Round 0.1 · original submission · Minor Revisions

This paper introduces a technique that using patterns of Zn enrichment to reflect differential bone development across ontogenetic stages. The methods have been well explained and results are promising. After minor revisions, mainly including some expression errors and expansion of description of patterns observed, the paper should be ready to be published.

**Language Note:** PeerJ staff have identified that the English language needs to be improved. When you prepare your next revision, please either (i) have a colleague who is proficient in English and familiar with the subject matter review your manuscript, or (ii) contact a professional editing service to review your manuscript. PeerJ can provide language editing services - you can contact us at [email protected] for pricing (be sure to provide your manuscript number and title). – PeerJ Staff

Reviewer 1 ·

Basic reporting

The writing is generally clear and professional, and maintains professional standards. See below for some minor recommended edits:
- The first sentence of the abstract should be re-worded, as its structure is a little counterintuitive. This is because the specimens being referred to are not the subjects of the sentence, and it is unclear what “the elemental composition” means. I suggest changing this to: “The elemental composition of seven specimens of Keichousaurus hui, a small pachypleurosaur (Reptilia: Sauropterygia) from the Middle Triassic of China, were analyzed via Micro X-ray fluorescence (Micro-XRF).”. I also don’t think you need the taxonomic information in parentheses, as this is already in the title.
- Line 267: I presume that “excessive” should be “excessively”.
- Line 275: “developmental” to “development”.
- Line 281: “a progressive progression” sounds clunky. Change to “demonstrates directionality”?
- Line 286: Insert “the” before “ossification”.
- Line 170: In most comparative anatomy literature, digit identity is assigned with numerals instead of number characters. I recommend changing “dt4” to dtIV.
- Line 290: Remove “bone” and the dtIV in parentheses. You have already defined it so you can simply use the abbreviation.

The references appear sufficiently broad to support the text and all data are shared. The study is self-contained and coherent.

Experimental design

The research is primary and fits with this journal’s stated guidelines. The methods are detailed and well explained. As another example of the utility of this method, I am sure the paper will be well-cited. Although I do not claim to fully understand the physics/chemistry of the method described, it appears obvious that the authors have gone to lengths to describe it sufficiently and include scan parameters in their supplemental data. These specimens are housed in a museum, so I presume that this experiment would be replicable. As far as I can tell, the use of these specimens was fully ethical.

Validity of the findings

Again, I have no experience using this method and I do not claim to be knowledgeable in its application, but the claims made by the authors seem rational to me. Having looked at the figures, I would come to the same conclusion about the distribution of Zinc as it related to our knowledge of amniote ossification patterns. These methods have been used before, on other specimens, but this paper does stand as an example of their heuristic power, and I am sure that it will be well-cited as a successful implementation. As a paleontologist, I can appreciate the value of studies that incorporate a high number of individuals (something that is rare in our field). This certainly adds to the studies value to both paleontologists, but also to developmental biologists with an interest in reconstructing developmental patterns in deep time. The field of biosignatures is quite active at the moment, and so I believe that the community will appreciate this paper. Sufficient data are available as supplements and are sound. The conclusions of the paper reflect its contents.

Additional comments

NA

·

Basic reporting

This is a very ingenuous study that links the presence of Zinc with the ontogenetic state in one species of plesiosaur. the results seems promising and this could be a technique that can change our way to determine the age of fossil individuals. I made a few comments on the pdf, but I also want to summarize some general comments I have about the way the results are presented:

1) Although the authors understand that mineralization can happen during the animal life and during fossilization, in one paragraph they seem to be mix these two. Make sure you clarify this.

2) The authors also understand the two types of ossification in vertebrates, but they discuss only endochondral ossification. I think that the paper would benefit of a more fine description of the patterns observed, while distinguish what type of bone they are referring to. The skull have elements that are endochondral in origin (e.g. braincase, jaws, epipterygoids), vs intermebranous ossification (parietals, frontals, premaxilla, etc). Including this distinction in their description can reveal even more detail. For instance, figure 2, the juvenile shows zinc in most of the skull, while figure 3 and 4, this is more concentrated in dermal elements. This can indicate that the braincase ad many endochondral origin elements are already ossiffied, while drama elements are still ongoing ossification in later stages.
Dr. Maisano did a lot work on ossification of reptiles, especially Squamates, the authors would benefit of looking at her papers, probably more than comparing reptiles with mice.
Maisano JA. 2001. A survey of state of ossification in neonatal squamates. Herpetological Monographs 15: 135-157.
Maisano JA. 2000. Postnatal skeletal development in squamates: its relationship to life history and potential phylogenetic informativeness. Ph D. thesis, Yale University, New Haven, CT, USA.
Maisano JA. 2002. Terminal fusions of skeletal elements as indicators of maturity in squamates. Journal of Vertebrate Paleontology 22: 268-275.

3) Images are fantastic, and the evidence seems clear, I would prefer if they include an statement of the preservation of bone elements, in this ways we can be certain that differences in Zinc concentration are due to ossification and not to element loss.

4) A trivial question, is it possible to determine what Zinc isotope is present? I know zinc has ben used indirectly to do do isotope dating.

Experimental design

Looks good, I just recommend including more detail about the types of ossification.

Validity of the findings

Original idea and impressive results

Additional comments

None

---

## Round 0.2 · accepted · Accept

A few suggestions have been made by the reviewer. See below:

"The second review of Skeletal ossification of Middle Triassic pachypleurosaur Keichousaurus hui (Reptilia: Sauropterygia) revealed by Zinc distribution has improved considerably. I considered the authors have tried to incorporate most of the reviewers comments. Although they responded well in the rebuttal letter, especially the distiction between endochondral and intermembranous ossification patters, and the limitations of this due to specimen preservation, perhaps this could be added somehow to the manuscript, especially for readers nor familiar with this fossils. "

The author may consider whether to add a few comments in the manuscript according to the reviewer's suggestion.

·

Basic reporting

The second review of Skeletal ossification of Middle Triassic pachypleurosaur Keichousaurus hui (Reptilia: Sauropterygia) revealed by Zinc distribution has improved considerably. I considered the authors have tried to incorporate most of the reviewers comments. Although they responded well in the rebuttal letter, especially the distiction between endochondral and intermembranous ossification patters, and the limitations of this due to specimen preservation, perhaps this could be added somehow to the manuscript, especially for readers nor familiar with this fossils. This of course if totally optional and the editor can opt for exclude this.

Experimental design

No comments on this part.

Validity of the findings

This seems to be a promising technique for studying of ossification of both living and fossil taxa.

Additional comments

No additional comments, great paper!